# Metabolomics and Cytokine Signatures in COVID-19: Uncovering Immunometabolism in Pathogenesis

**DOI:** 10.3390/metabo15090608

**Published:** 2025-09-11

**Authors:** Mohammad Mehdi Banoei, Abdulrazagh Hashemi Shahraki, Kayo Santos, Gregory Holt, Mehdi Mirsaeidi

**Affiliations:** 1Department of Critical Care Medicine, University of Calgary, Calgary, AB T2N 4Z6, Canada; mmbanoei@ucalgary.ca; 2Department of Biomedical Engineering, Schulich School of Engineering, University of Calgary, Calgary, AB T2N 1N4, Canada; 3Division of Pulmonary, Critical Care, and Sleep, College of Medicine—Jacksonville, University of Florida, Jacksonville, FL 32209, USA; hashemishahrak.a@ufl.edu; 4Division of Pulmonary and Critical Care, University of Miami, Miami, FL 33146, USA; kayohenrique.md@gmail.com (K.S.); gholt@miami.edu (G.H.)

**Keywords:** metabolic biosignatures, cytokines biosignatures, immunometabolism, COVID-19, metabolites-cytokine correlation

## Abstract

Background: This study aimed to analyze metabolic changes in blood samples from patients with confirmed COVID-19 to explore the correlation between metabolomics and cytokines in survivors and non-survivors of SARS-CoV-2 infection. Understanding the complex biochemical and immunometabolic mechanisms underlying SARS-CoV-2 infection is essential for elucidating the pathophysiology and virulence of COVID-19. Methods: This study included 40 hospitalized COVID-19 patients and 40 healthy controls. Serum metabolic profiles were analyzed using ultra-high-pressure liquid chromatography-mass spectrometry (UHPLC-MS), and cytokine levels were measured using ELISA. Results: Our study defined three clear metabolic phenotypes among survivors and non-survivors of COVID-19 compared with healthy controls, which might be related to mortality, severity, and disease burden. A strong relationship was observed between certain inflammatory markers, including IL-1β, IL-2, IFN-β, IFN-γ, IL-17, and GM-CSF, as well as several metabolites, particularly in COVID-19 non-survivors, such as LysoPCs, 3-hydroxykynurenine, and serotonin. Different metabolite-cytokine correlation patterns were observed according to patient outcomes, indicating unique correlations between metabolic and immune responses in survivors and non-survivors. Metabolic phenotypes were associated with clinical outcomes, comorbidities, and sex-related differences. Kynurenine and related metabolites of tryptophan metabolism were closely correlated with COVID-19 severity, age, and mortality. Compared with survivors and healthy controls, non-survivors displayed higher IL-6, together with distinct metabolic changes. These included increased kynurenine through the IDO1 pathway, elevated glucose and lactate reflecting hyperglycolysis and energy stress, and higher xanthosine from purine turnover. Stronger cytokine–metabolite correlations in this group point to tightly linked immunometabolic activation. Conclusions: Metabolomic profiling revealed distinct metabolic phenotypes that could be associated with the severity and inflammation levels of COVID-19. Correlation analysis between metabolites and cytokines demonstrated strong intercorrelations between specific metabolites and cytokines, indicating a strong interrelationship between inflammatory markers and metabolic alterations. Specific metabolic pathways associated with cytokines and their clinical relevance may serve as potential therapeutic targets.

## 1. Introduction

The COVID-19 pandemic has seriously affected both global health and economies, causing over 778 million infections and more than 7.1 million deaths worldwide, as of June 2025 [1]. The mechanisms by which SARS-CoV-2 may affect the human body include direct injury from viral cytotoxicity, endothelial damage through angiotensin-converting enzyme 2 (ACE-2) receptor-mediated viral entry, an extreme immune response, and virus-independent mechanisms such as unknown tissue-specific effects and immunopathology [2]. Although research into COVID-19 immunometabolism has expanded, several central questions remain unanswered. First, the causal links and cellular origins of cytokine–metabolite networks such as the tryptophan/kynurenine and arginine pathways are still unclear [3,4]. Second, it is not well established how these networks shift from the acute stage of disease into recovery or Long COVID, or whether immune-directed therapies normalize the disrupted metabolism [3,5]. Third, the influence of obesity and related metabolic conditions on inflammatory metabolism needs a clearer mechanistic explanation [6]. Fourth, the extent to which viral variants and vaccination shape distinctive immunometabolic patterns is uncertain. Finally, it remains to be seen if combined cytokine and metabolite signatures can reliably predict outcomes across independent cohorts [7]. The elaborate interaction between the host immune response and metabolic changes partly explains the mechanisms behind the cytokine storm and multiorgan dysfunction in severe cases. Severe COVID-19 cytokine storm develops through a mix of immune over-activation, altered metabolism, and breakdown of normal control mechanisms. When viral components are detected by pattern recognition receptors such as TLRs, macrophages and dendritic cells release IL-6, TNF-α, and IL-1β. Instead of settling down, these signals are amplified by PAMPs, DAMPs, and inflammatory cell death, like pyroptosis or necroptosis [8]. The result is a self-reinforcing cycle that drives widespread inflammation. High levels of cytokines then weaken endothelial and epithelial barriers, raise vascular permeability, trigger clotting pathways, and promote microthrombi. Damage spreads across the lungs, heart, kidneys, liver, and brain [9]. Metabolic rewiring is deeply involved. Indoleamine 2,3-dioxygenase (IDO1) activation drains tryptophan and builds up kynurenine, which dampens T-cell activity but fuels AhR-driven pro-inflammatory signaling. Disturbed arginine metabolism limits nitric oxide, undermining endothelial function and encouraging clot formation. A glycolytic shift in immune cells, the Warburg effect, boosts lactate production and feeds back into cytokine release [10,11]. Alongside this, depletion of glutathione, accumulation of ROS, and mitochondrial failure intensify tissue injury. When regulatory brakes like IL-10 signaling fail, the immune system cannot contract, and unchecked inflammation advances toward multi-organ failure [12].

A dysregulated immune response in severe cases of infection characterizes COVID-19 pathophysiology. The elevation of systemic levels of pro-inflammatory cytokines/chemokines characterizes severe clinical disease and is associated with poor clinical outcomes [13]. Significantly higher levels of inflammatory cytokines and inflammatory markers, such as IL-6, IL-8, IL-10, and C-reactive protein and lactate dehydrogenase (LDH), are associated with the critical group than with the moderate group [14,15]. Multiple cohorts and meta-analyses show that circulating cytokines and inflammatory markers consistently rise with worsening COVID-19. Elevated IL-6, IL-8, and IL-10 at the time of admission predict progression, ICU transfer, and mortality, even when adjusted for other variables [16]. CRP is also markedly higher in severe cases compared to moderate disease and shows strong predictive value for adverse outcomes in both individual studies and quantitative reviews [17]. LDH follows the same pattern, linking closely with severity, ICU requirement, and risk of death, with meta-analyses confirming its rise in severe or critical groups [18]. Taken together, these findings align with our observation that IL-6, IL-8, IL-10, CRP, and LDH are enriched in critical illness and offer prognostic insight beyond the initial clinical presentation [19].

SARS-CoV-2 induces a huge metabolic change that alters the energy pathways of immune cells. Typically, cells of the immune system gain energy through oxidative phosphorylation. Metabolic programming is a hallmark of immune cell activation that controls effector functions and fate decisions. Environmental signals such as antigens, cytokines, hypoxia, and nutrients act through mTOR, HIF-1α, and AMPK to reprogram glycolysis, the TCA cycle, fatty acid oxidation, and oxidative phosphorylation, shaping immune cell activity. Naïve T cells depend on oxidative phosphorylation and fatty acid oxidation but shift to aerobic glycolysis after activation to sustain biomass growth and cytokine release. Memory T cells return to mitochondrial metabolism and fatty acid oxidation [20,21]. Lineage specification is also guided by metabolism: Th17 cells rely on glycolysis and glutaminolysis under HIF-1α and Myc control, while Tregs depend on fatty acid oxidation and mitochondrial pathways [20]. In myeloid cells, M1 macrophages are glycolytic and accumulate succinate, which stabilizes HIF-1α to promote IL-1β. In contrast, the metabolite itaconate, produced by IRG1/ACOD1, limits inflammation by inhibiting SDH and activating Nrf2 [22,23,24]. In COVID-19, monocytes display a HIF-1α–driven glycolytic shift with elevated lactate that amplifies cytokine production [10]. Plasma kynurenine from IDO1-mediated tryptophan breakdown correlates with IL-6 levels and disease severity [11]. Disturbed arginine and nitric oxide metabolism further disrupts endothelial function and host defense [25]. These metabolic programs directly regulate cytokine production, antigen presentation, proliferation, cytotoxicity, and lineage fate rather than serving only as correlates [20].

The severity of the disease is attributed to the significant metabolic changes reported in COVID-19 [26,27]. The first reports have shown alterations in carbohydrate metabolism, aminotransferase activity, lipid metabolism, and amino acid turnover, which may be events pertinent to immune responses [27]. For instance, a hypermetabolic state characterized by increased glycolysis usually exists during severe COVID-19 cases that meet the enhanced energy and biosynthetic needs of activated immune cells [10]. Similarly, lipid metabolism abnormalities among COVID-19 patients may affect immune signaling, membrane-bound protein functionality, and cell membrane flexibility [28]. Severe COVID-19 is marked by broad metabolic remodeling. Carbohydrate pathways shift toward hyperglycolysis, with HIF-1α–driven monocyte reprogramming and increased lactate that fuels inflammatory signaling [10]. Aminotransferase activity often rises, and a recent systematic review and meta-analysis reported that both the De Ritis ratio (AST/ALT) and elevated AST or ALT are linked to greater severity and higher mortality [29]. Lipid handling is also disrupted, with altered transport and plasma lipid signatures that track with disease severity in cross-omic studies [30]. Amino acid metabolism changes as well: activation of the tryptophan to kynurenine pathway aligns with IL-6 levels and clinical status, while disrupted arginine and nitric oxide pathways impair endothelial and immune function [11,25]. Collectively, these alterations reveal a recurring pattern of metabolic rewiring that parallels escalating inflammation and poor outcomes. Other metabolites, such as kynurenine, have been linked to immunosuppression in COVID-19 and may influence its course [11]. Mapping virally induced metabolic perturbations is important for understanding how viruses meet their biosynthetic demands. This information offers a new avenue for controlling SARS-CoV-2 infections. Furthermore, how metabolic interplay and immunity-versus-virally induced metabolic persuasions offer prospects for predicting trajectories of COVID-19 infection and identifying people who are likely to have worse outcomes. Biological pathways can play different roles in the induction of pro-inflammatory mechanisms during COVID-19 infection [31]. We aimed to determine the serum metabolomic profile of patients with COVID-19, including survivors and non-survivors, and their correlation with the cytokine profiles. We hypothesized that these correlations would index a coordinated immunometabolic state because inflammatory cytokines actively reprogram metabolism and metabolites reciprocally modulate cytokine output; accordingly, we treat correlations as composite readouts of disease activity rather than causal claims.

## 2. Methods and Materials

### 2.1. Patients’ Enrollment

This study included 40 hospitalized and PCR-positive COVID-19 patients; some were survivors (*n* = 20), whereas others were non-survivors (*n* = 20). Data were obtained from the University of Miami Hospital, Miller School of Medicine, Miami, FL, USA, for patients admitted between June 2020 and April 2021. Serum samples were obtained within the first two days following the onset of illness. More than 100 clinical comorbidities and common variables related to biochemical standards have been collected for COVID-19 patients. Correlation analysis was performed on more than twenty-one demographic and clinical data points. Additionally, a control group of 40 age- and sex-matched healthy participants from the University of Miami Hospital was included. This study was approved by the institutional review board of the University of Miami. Blood samples were collected following receipt of informed consent from the patient or a family member.

### 2.2. Liquid Chromatography-Mass Spectroscopy (LC-MS)-Based Metabolomics Analysis

50 μL of serum samples were used for the semi-non-targeted metabolomic analysis using a Q Exactive HF Hybrid Quadrupole-Orbitrap Mass Spectrometer (Thermo-Fisher, Calgary, AB, Canada). Chromatography was performed using a 2.1 mm × 100 mm long Synchronism HILIC (Thermofisher, Calgary, AB, Canada) LC column packed in-house with 3 µm porous Hyperarc particles. We identified and quantified 166 metabolites in this analysis. Maven software V.12, an open-source software, was used for processing metabolomics data obtained by LC-MS [32]. All procedures for sample preparation, ultra-high-pressure liquid chromatography-mass spectrometry (UHPLC-MS, Thermo-Fisher, Calgary, AB, Canada) data acquisition and processing, and statistical analysis are detailed in our recently published article [33].

### 2.3. Cytokines Analysis

Fourteen cytokines, including TNF-α, IL-6, IL-2, IFN-γ, IL-1β, IL-17A (CTLA-8), IL12/IL-23p40, IL-7, IL-2R, IFN-α, IL-8 (CXCL8), IFN-β, GM-CSF, and IL-10, were measured in patients with COVID-19 using ELISA kit from R&D system@ Rat Luminex^®^ Discovery Assay (Catalog #: LXSARM) Minneapolis, MN, USA.

### 2.4. Data Analysis

Metabolite identifications and quantification were performed using El-MAVEN V.12 (Elucidata Inc., San Francisco, CA, USA) to measure the ion intensities of compounds [34]. Ion peaks were selected based on the mass-to-charge ratio (*M*/*Z*), retention time (RT), and ion intensity of metabolites compared to the pre-and post-blank.

For data analysis, principal component analysis (PCA) was applied to have an overview and find outliers in an unsupervised manner. Partial Least Squares-Discriminant Analysis (PLS-DA) was used to discriminate the phenotypes between two or more groups. A non-parametric analysis of variance was used to detect differences in raw ion intensities among samples between different groups. FDR correction by Benjamini–Hochberg test followed by Bonferroni corrections.

Student’s *t*-test was performed to indicate differences between samples from the two groups. All statistical tests were two-sided, and adjusted *p*-values below 0.05 were considered statistically significant. Univariate analysis identified significantly altered metabolites among three cohorts, which were selected for categorizing metabolic phenotypes. Correlations among variables of interest were performed using the Spearman correlation test, and when necessary, corrected for multiple inferences using Holm’s method. Power analysis was performed to calculate the minimum number of samples required to detect a statistically significant difference between two populations, based on a user-specified degree of confidence. MetaboAnalyst 6.0 [35], GraphPad Prism 9.5.1 and SIMCA P v 14.0 were used for comprehensive metabolomics data analysis.

## 3. Results

### 3.1. Patient Characteristics

The patient population for this retrospective observational study consisted of 40 patients admitted to the hospital with COVID-19 and a group of 40 age- and sex-matched healthy controls. The COVID-19 cohort included 20 survivors and 20 non-survivors with a mean age of 73.3 ± 11.4 and 70 ± 12.5, respectively. There was no significant difference in sex (*p* = 1.000) or age distribution between males and females (*p* = 0.549) (Table 1).

### 3.2. Cytokine Profiling of COVID Patients

Appendix A shows the analysis of 14 cytokines in the non-survivors and survivors of COVID-19. IL-6, TNF-α, IFN-γ, and IL-2 were significantly different between the two groups with a significant FDR (*p* < 0.05) for IL-6 and TNF-α; IFN-β, IL-2R, IL-10, and IL-8 also had FC > 1.5 or <1.5 between the groups (Appendix A). Total serum immunoglobulin G (IgG) mean (± SD) was measured at 1.87 (±1.70) among COVID patients. IgG levels were also not significantly different (*p* > 0.05) between non-survivors and survivors, with a mean of 2.11 (±1.74) and 1.78 (±1.70), respectively.

### 3.3. Metabolomic Profiling Showed Three Main Metabolic Phenotypes in Patients with COVID-19 Compared with Those in HCs

Three main metabolic phenotypes were identified by comparing COVID-19 non-survivors and survivors with those of healthy controls using a *t*-test analysis (Appendix A). All three phenotypes were illustrated using multivariate analysis via PLS-DA (Figure 1) and hierarchical analysis (heatmap) (Figure 2). Patients are not restricted to one phenotype, since each sample can carry metabolites from all three patterns. Metabolic phenotype 1 exhibited a significant change in the concentrations of metabolites among COVID-19 non-survivors compared to COVID-19 survivors and HCs, and a significant difference between COVID-19 survivors and HCs, suggesting a possible correlation of these metabolites with the severity or increased inflammation of COVID-19 infection. Phenotype 1 comprises 14 metabolites, including kynurenine, xanthine, 3-ureidopropionate, D-glucuronic acid, uridine, acetylspermine, and 5-hydroxy-L-tryptophan (Appendix A). Examples include kynurenine from the tryptophan–kynurenine pathway, acetylspermidine from polyamine turnover, and 5-hydroxytryptophan as a serotonin precursor. Boxplots in Figure 2B illustrate the stepwise rise across groups with exact *p*-values noted, and representative chromatograms are shown in Appendix A. The pattern reflects IDO1/AhR-driven inflammatory catabolism along with enhanced polyamine acetylation in fatal cases. Phenotype 2 represents metabolites significantly altered exclusively in COVID-19 non-survivors, while their concentrations showed no significant differences between COVID-19 survivors and healthy controls. This indicates that metabolic phenotype 2 can be linked to poorer (mortality) outcomes and is a significant indicator of the severe progression of COVID-19. This metabolic phenotype included 17 metabolites, such as xanthosine, glucose, malic acid, and N-formylglycine. Carbohydrate and purine metabolism shift with disease severity. Glucose is most elevated in non-survivors, xanthosine from purine breakdown also rises in this group, while phenylacetylglutamine (PAGln) is comparatively higher in survivors. These findings point to hyperglycolytic stress and accelerated nucleotide turnover in the poorest outcomes, with survivors showing a distinct gut–liver axis signal through PAGln. Boxplots and chromatograms are presented in Figure 2B and Appendix A. Phenotype 3 is characterized by metabolites with similar concentrations in both COVID-19 non-survivors and survivors but significantly altered compared to the HC group, indicating COVID-19 infection-related metabolite alterations irrespective of disease severity. The higher number of metabolites in phenotype 3 confirmed the significant effect of the disease on metabolism. Metabolites such as indole-3-acetic acid, azelaic acid, suberic acid, glutarate, guanidinoacetate, and 3-hydroxykynurenine are categorized under this phenotype. This module links oxidative kynurenine pathway activity with immunoregulatory signals. 3-hydroxykynurenine is elevated in non-survivors, azelaic acid rises in both COVID groups compared with healthy controls, and itaconate, a macrophage-derived metabolite, is higher in COVID-19 with a trend toward greater levels in survivors, suggesting a more active counter-inflammatory response. Group differences are shown in the boxplots of Figure 2B, with chromatograms in Appendix A.

### 3.4. Correlation Between Metabolic Phenotypes 1–3 and Cytokine Profiling

An overview of the correlation between all metabolites and cytokines is available in the Appendix A. Regarding metabolic phenotype 1, the relationship between kynurenine and various cytokines differed between survivors and non-survivors. Kynurenine was strongly correlated with IL-10, IL-8, IL-7, IL-6, IL-2R, IL-12, and IFN-α levels in patients with COVID-19. In particular, IL-12 and IFN-α were more strongly related to L-cystathionine, kynurenine, 5-hydroxy-L-tryptophan, uridine, and D-glucuronic acid in non-survivors. In addition, nearly all metabolites of phenotype 1 were positively correlated with IL-8 levels among survivors (Figure 3). The results showed that IL-1β, IL-2, IFN-β, IFN-γ, IL-17, GM-CSF, and TFN-α had a slightly higher correlation with metabolic phenotype 2 in non-survivors, indicating a potential association with COVID-19 mortality outcome. For metabolic phenotype 2, C10, C5OH, phenylacetylglutamine, and N-amidino-L-aspartate showed higher correlation coefficients with IL-12 and IFN-α in non-survivors; however, methyl malonate showed a stronger cytokine correlation in survivors (Figure 3). Further analysis showed an overall lower correlation between the cytokines IL-2R, IL-6, IL-7, IL-8, and IL-10 and metabolic phenotype 3 among non-survivors, indicating a lower association of these cytokines with COVID-19-related metabolites. In metabolic phenotype 3, L-arginine, 6-carboxyhexanoate, and L-glutamic acid showed lower cytokine correlation coefficients among non-survivors. In general, a few metabolites were strongly associated with IL-12, IFN-γ, TNF-α, GM-CSF, IFN-β, and IFN-α in the survivors (Figure 3). Figure 4 shows some key examples of important metabolite-cytokine correlations (*p* < 0.05) that appeared almost entirely within non-survivors, except for the itaconate and IL-1β correlations, which were significant in survivors. A significant (*p* < 0.05) correlation among metabolites within metabolic phenotypes between non-survivors and survivors is shown in Appendix A.

Further investigation showed that IL-8, IL-7, and IL-2R levels were less highly correlated with other cytokines in survivors. In contrast, IL-2R, IL-6, IL-8, and IL-10 levels were highly correlated in non-survivors compared to those in survivors (Appendix A).

### 3.5. Metabolomics and Patients’ Demographics, Clinical Data, and Comorbidities

Further investigation demonstrated a strong correlation between metabolic phenotype 1 (intensity of severity-related metabolites) and clinical data, comorbidities, and outcomes. Metabolic phenotype 1 metabolites were highly correlated with COVID-19 outcomes such as mortality, ARDS, and ICU admission. These metabolites also correlated more strongly with the *P*/*F* ratio, respiratory rate > 24, intubation, and pneumonia. Females showed a higher correlation with most of this metabolic phenotype than did males (Figure 5). Metabolic phenotype 2, some metabolites, such as C10:1, C5OH, phenylacetyl glutamate, xanthosine, L-isoleucine, and N-formylglycine, were mostly correlated with poor outcomes. Interestingly, glucose, allose, and 2-oxobutanoate levels were highly associated with obesity. Among the metabolic phenotype 3, N4-acetylcytidine, C-glycosyl-tryptophan, and kynurenic acid were mainly correlated with COVID-19 outcomes and comorbidities. Interestingly, the respiratory rate > 24 was negatively correlated with most of the metabolites of this phenotype. IL-6 was the only cytokine that showed a positive correlation with mortality, ARDS, and ICU admission outcomes in the investigated cohort (Figure 5). All cytokines showed high correlations with INR and PT, history of infectious disease, and obesity. Females had higher cytokine concentrations than males. Higher correlations between cytokines and metabolic phenotype 1 among females may indicate a higher level of inflammatory mechanisms among female patients. Kynurenine, the most well-known metabolite related to inflammatory mechanisms, had a higher correlation with COVID-19 outcomes and age > 65 years, along with two other metabolites from tryptophan metabolism, 5-hydroxy-l-tryptophan and kynurenic acid (Figure 5).

## 4. Discussion

We observed that elevated levels of several metabolites, including urate, 5-hydroxyisourate, guanosine, kynurenine, inosine, pyruvate, LysoPCs, 3-hydroxykynurenine, and serotonin, were strongly associated with poor clinical outcomes and increased serum inflammatory cytokine levels in patients with COVID-19. Notably, IL-2R, IL-6, IL-7, IL-8, and IL-10 showed lower correlations with the metabolites for the worst outcomes, indicating that these cytokines might have different regulatory mechanisms during severe COVID-19. We also observed a strong association between metabolic phenotypes and clinical outcomes, noting that phenotype 1 metabolites were strongly correlated with mortality, ARDS, severity, and clinical parameters such as increased respiratory rates and lower *P*/*F* ratios. SARS-CoV-2 is associated with an increase in pro-inflammatory cytokines, which may cause cytokine storm and hyperinflammation in patients with severe COVID-19. The cytokine storms are characterized by significant induction of specific cytokines such as IL-6, IL-2, IL-7, GM-CSF, and IFN-γ in COVID-19 patients, which are different from other respiratory viruses with the increase in cytokines such as IL-2, IL-10, IL-4, or IL-5 [2]. In severe COVID-19, elevated levels of cytokines such as TNF-α, IL-6, IL-8, and IL-10 have been significantly associated with a reduction in T-cell counts [2]. However, IL-6, a significant cytokine in COVID-19, is mainly associated with two JAK/STAT pathways [36]. Our data showed that most metabolites had a lower correlation with IL-6 than other cytokines such as IL-2, IFN-β, and GM-CSF. This may describe the role of different SARS-CoV-2 variants in the induction of inflammatory response [37] and/or different stages of the disease or the presence of potential comorbidities in the group. The cytokine–metabolite relationships we report are associative rather than causal. We view these correlations as integrated markers of ongoing immunometabolic activity that may aid risk stratification, while acknowledging that causal links will need to be tested in longitudinal and perturbational studies.

Several studies have shown a correlation between metabolic changes and cytokine expression in COVID-19 patients. Importantly, it has been demonstrated that some metabolic agitation in COVID-19 is mediated by elevated pro-inflammatory cytokines, oxidative stress, and deregulation of the renin–angiotensin–aldosterone system in many tissues, indicating that metabolic and cytokine changes are key contributors to tissue dysfunction [38]. The metabolic machinery required for the replication of SARS-CoV-2 is characterized by metabolic perturbations, such as alterations in nucleic acid pathways and depletion of malic acid and GMP from host cells [39], which contribute to the production of the viral capsid through the mobilization of free fatty acids [40]. SARS-CoV-2 infection directly affects cells and tissues via metabolite perturbations, which may influence the host’s inflammatory responses [41]. On the other hand, dysregulated immunometabolism causes immune dysregulation and cytokine storm in patients with COVID-19 in severe form. For instance, upregulation of glycolysis and glutaminolysis can lead to hyperinflammatory responses, tissue damage and multi-organ dysfunction. Moreover, the interplay between immune cells and viruses for metabolic competition may shape host–pathogen interactions, contributing to disease development. This indicates that metabolic crosstalk between immune cells and SARS-Cov-2 may be important for understanding new avenues of targeted interventions that aim to modulate host immune responses [8].

Identifiable metabolic alterations include essential metabolic pathways related to amino acid and energy metabolism in COVID-19 patients. Among these metabolites, L-tryptophan and L-kynurenine play key roles in COVID-19 pathogenesis [4,42]. Our metabolomic study showed that several metabolites in tryptophan metabolism or the tryptophan-kynurenine pathway differentially changed depending on the severity of COVID-19.

It has been reported that metabolites, such as agmatine, putrescine, and 2-quinolinecarboxylate, are involved in the activation of the NF-kB pathway, thereby promoting an increase in the secretion of the pro-inflammatory cytokines TNF-α and IL-6, which are two important cytokines governing the severity of COVID-19. Longitudinal studies have shown that cytokines such as IL-6 and IL-10 can also be used as markers of liver injury severity in patients with COVID-19 [43].

It has been shown that IL-4 is not only associated with liver repair but is also suggested to be a restoring factor of liver function. This relationship between cytokine profiles and metabolome changes indicates the complexity of recovery from COVID-19 [44]. The quantification of cytokines in plasma revealed associations with antibody decay in COVID-19 convalescent patients for cytokines, such as M-CSF and IL-12p40, and metabolites, such as glycylproline and long-chain acylcarnitines. Glycylproline, a product of dipeptidyl peptidase 4 activity, has recently been shown to provide a basis for the rapid waning of SARS-CoV-2-specific antibodies. An experiment that entailed supplementing glycylproline in experiments with SARS-CoV-2 vaccination in healthy mice resulted in a down-regulation of SARS-CoV-2-specific antibody levels and a suppressed immune response [45]. Major metabolites of energy, amino acids, and lipid metabolisms were differentially altered and closely related to inflammatory responses and disease progression. L-arginine participates in pathways related to inflammation, immune regulation, and NO generation, which are necessary for the differentiation, survival, and proliferation of Th cells. IL-18, IL-1β, and IL-23 have been correlated with arginine biosynthesis in COVID-19 [46]. Our study showed L-arginine had a higher correlation with most cytokines in survivors than in non-survivors. Our study also showed that glucose and allose, metabolites associated with mortality, were strongly correlated with TNF-α, GM-CSF, IFN-γ, IL-17, IFN-β, IL-2, and IL-1β levels in non-survivors. Profiling serum samples from COVID-19 patients of varying severity showed strong correlations between metabolites and pro-inflammatory cytokines/chemokines, including IL-6 and IL-1β, highlighting a regulatory interplay between arginine, tryptophan, purine metabolism, and hyperinflammation [4]. This study showed that the levels of classical pro-inflammatory cytokines such as IL-6, IP-10, and M-CSF increased during hospitalization. Anti-inflammatories such as IL-10 and IFN-α2 also increased over the same duration, suggesting a protective immune response. IDO1 inhibition has been demonstrated to suppress SARS-CoV-2-induced pro-inflammatory cytokine release by reducing the activity of tryptophan-metabolizing enzymes. Conversely, inhibition of IMPDH, a key enzyme in purine metabolism, during hyperinflammation results in the upregulation of several pro-inflammatory cytokines, including IL-6 and IL-1β [4]. Significant collective data have revealed strong correlations between arginine metabolism and key pro-inflammatory cytokines expressed during this process, including GM-CSF, IFN-γ, IL-17A, IFN-β, IL-2, and IL-1β [4]. Metabolic phenotyping revealed a systemic reaction and immunometabolic disorder associated with the infection. The tryptophan pathway is linked to neuroinflammation and neuropsychiatric consequences [47]. In male patients, the concentration of kynurenic acid and KA/kynurenine ratio were positively influenced by age and inflammatory cytokines and chemokines. This indicates that KA has a sex-specific link with the immune and clinical results of COVID-19 [48]. Our study also demonstrated the association between KA and age > 65 years, male sex, ARDS, and severity. An NMR-based study showed correlations among plasma cytokines, lipoproteins, and metabolites in distinct groups. Patients in the recovery phase demonstrated an incomplete metabolic recovery postinfection [49].

IL-10 and IL-6, cytokines indicative of inflammation, showed a significant negative correlation with glutamine, a key metabolic marker of COVID-19 infection. Increased levels of IL-6 and IL-8 were correlated with acute-phase proteins (CRP and ferritin), glycoproteins (Glycs), and ketone bodies, highlighting the connection between the immune response and metabolic disturbances. IFN-α2 levels negatively correlated with inflammatory Glycs and the supramolecular phospholipid composite (SPC) and negatively correlated with all measured cytokines. MCP1 was positively correlated with all major lipid and HDL parameters, suggesting its unique role in lipid metabolism in COVID-19. Significant negative correlations between IL-6 and ApoA1, particularly in the dense HDL subfraction (H4A1), indicated inflammation-induced changes in the lipoprotein profiles [50]. Our study showed that LysoPCs correlated more strongly with cytokines, notably GM-CSF and IL-1β.

Tocilizumab, an IL-6R inhibitor, only partially reverses the metabolic consequences of severe COVID-19. This confirms that immune mediators drive metabolic alterations upon infection and suggests a dynamic interplay between the immune system and metabolism. The activities of particular cytokines and different immune cells have been proposed to modulate metabolic pathways centered on lipid metabolism and energy use [51].

Plasma metabolomics and cytokine profiling in critically ill patients with SARS-CoV-2 infection revealed four metabolites (3-hydroxybutyrate, lactate, leucine, and phenylalanine) and five cytokines/growth factors (CXCL9, CXCL10, HGF, IL-6, and SCF) whose concentrations were associated with a favorable patient outcome [52].

Previous studies have shown alterations in metabolic profiles with simultaneous changes in immune cell functions, including implications for COVID-19 severity and progression [53,54]. An interplay between immune responses and metabolic pathways is also apparent in pregnant women with COVID-19, where different disease severities generate distinct immune-metabolic profiles between mothers and neonates [55].

The correlations identified point to linked immune and metabolic activity rather than independent signals. Inflammatory cytokines such as IL-6 and IFN pathways can redirect metabolism toward routes like tryptophan to kynurenine or glycolysis to lactate, while certain metabolites in turn influence cytokine output and vascular or immune function. These paired readouts therefore capture integrated features of disease biology. Clinically, such signatures have several applications: they can sharpen prognostic stratification when added to standard labs and covariates, help define immunometabolic subtypes such as IL-6–kynurenine–driven versus glycolysis/lactate–driven patterns that may guide therapy or trial design and serve as markers of pharmacodynamic response to anti-inflammatory or supportive treatments. Although correlation does not establish causation, the biological logic of these axes makes them suitable for targeted validation in independent cohorts and longitudinal studies.

We found that cytokine–metabolite correlations were stronger in females. This is biologically plausible for several reasons. Hormonal influences play a role: estrogens tend to boost innate antiviral and type I IFN responses as well as humoral immunity, while androgens can shape ACE2/TMPRSS2 expression and inflammatory pathways. X-linked immune genes such as TLR7 may also contribute, since gene-dose or escape effects can heighten IFN signaling in some women. In addition, sex-related metabolic remodeling has been reported in COVID-19, spanning lipid, pentose, bile acid, and aromatic amino acid/tryptophan pathways. Prior studies further show sex-dependent differences in the kynurenine axis, including distinct associations of kynurenic acid by sex, which emphasizes that immunometabolic coupling does not operate identically in males and females. We interpret the stronger correlations in females as associative evidence of tighter immunometabolic coordination, while recognizing the limitations of sample size, treatment and variant heterogeneity, and the need for independent validation [56,57,58].

Patient enrollment (2020–2021) took place across several SARS-CoV-2 variant waves, including Alpha and Delta. Viral genotyping was not available, so analyses were not stratified by variant. Since variants differ in both immune and metabolic responses, and clinical management changed during this period, the applicability of our immunometabolic signatures to specific variant contexts, particularly the Omicron era, is limited. Future work should include viral sequencing or reliable proxy assignment and conduct variant-stratified analyses. Also, patients were admitted at different points in their illness, our baseline reflects an admission-time snapshot rather than a consistent disease day. This heterogeneity may blur or weaken stage-specific effects. Studies that stratify by illness duration and include serial sampling will be important to confirm these observations. In-hospital therapies such as glucocorticoids, antivirals, IL-6 receptor blockers, anticoagulants, and glucose-lowering agents can rapidly influence cytokine and metabolic profiles. Detailed information on timing and dosing at the time of sampling was not consistently available, and the cohort size did not allow for robust adjustment. Residual confounding is therefore likely. The profiles presented here should be viewed as reflecting both disease activity and concurrent treatment, and future work will need to include more granular medication data and stratified analyses

The limitations of this study include the small sample size. Also, this study was a modest single-centre cohort; both power and generalizability are limited. Even with FDR correction, some results may be unstable and will need confirmation in larger multi-centre studies. In addition, other limitations include a lack of comparative samples across different severities of COVID-19 (mild and moderate), limited representation of SARS-CoV-2 variants, a lack of comparisons between hospitalized and outpatient cases, and limited number of cytokines. Longitudinal data could help determine whether the observed changes in metabolites and cytokines persisted or evolved during the disease. Investigations of the role of confounding factors, such as age, sex, comorbidities, and medications, have not been fully considered in this study. Finally, a separate validation cohort was required to verify the findings.

In conclusion, we identified a distinct profile of metabolites that may serve as novel biomarkers for poor outcomes in COVID-19. These findings provide valuable insights into disease severity, though further studies are necessary to validate and expand upon these results.

## Figures and Tables

**Figure 1 metabolites-15-00608-f001:**
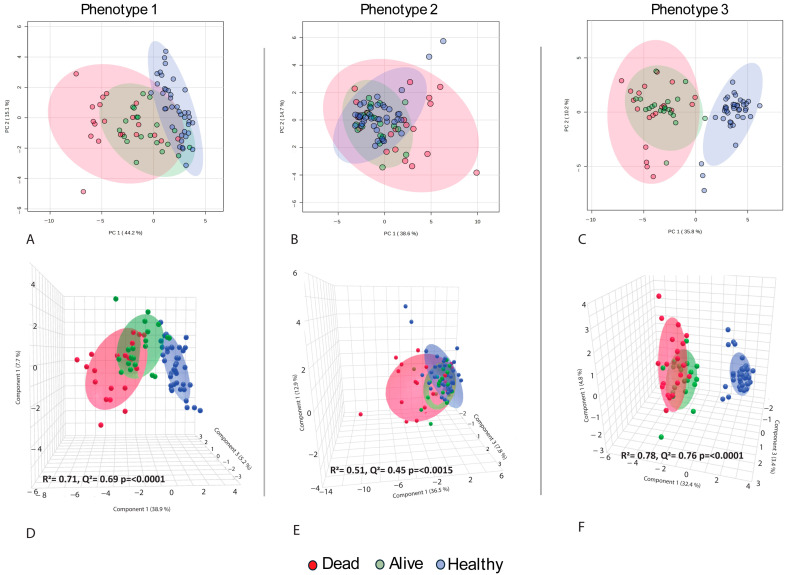
(**A**–**C**): PCA demonstrates the differentiation of the three cohorts based on phenotypes 1 to 3. (**D**–**F**): The PLS-DA using metabolic phenotypes highlights distinct changes in metabolites across the cohorts based on phenotypes 1–3. The models showed Phenotype 1: R^2^ = 0.71, Q^2^ = 0.69; Phenotype 2: R^2^ = 0.51, Q^2^ = 0.45; Phenotype 3: R^2^ = 0.78, Q^2^ = 0.76. Predictive ability is high for Phenotypes 1 and 3 (Q^2^ > 0.6) and moderate for Phenotype 2. The close agreement of R^2^ and Q^2^ suggests no evident overfitting.

**Figure 2 metabolites-15-00608-f002:**
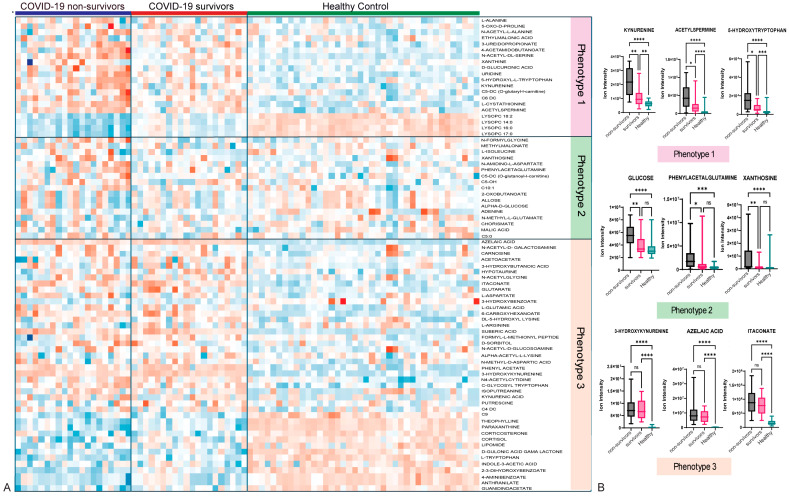
Heatmap analysis illustrates the metabolite variations associated with different metabolic phenotypes across three groups: COVID-19 non-survivors, COVID-19 survivors, and healthy controls (**A**). Examples of metabolites related to each metabolic phenotype are shown (**B**). Statistical significance was determined using. * *p* < 0.05; ** *p* < 0.01; *** *p* < 0.001; **** *p* < 0.0001; ns: non significant.

**Figure 3 metabolites-15-00608-f003:**
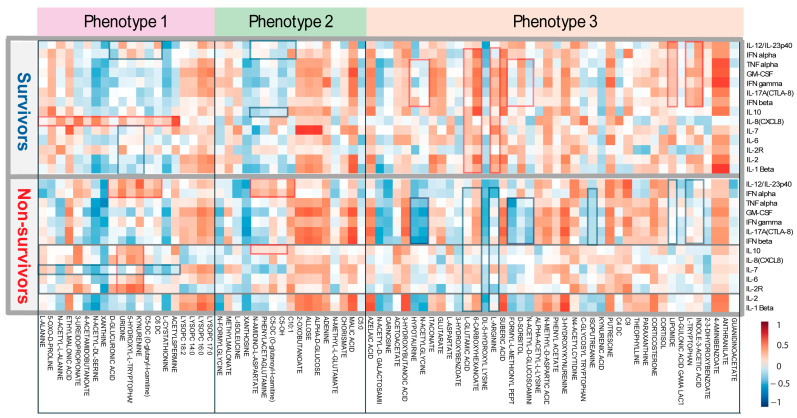
Correlation analysis reveals the correlation coefficients between metabolites from three metabolic phenotypes and 14 cytokines of COVID-19 survivors and non-survivors. The figure highlights significant differences in metabolite-cytokine correlations between the non-survivor and survivor cohorts. Key differences are marked by red rectangles (indicating higher correlations) and blue rectangles (indicating lower correlations).

**Figure 4 metabolites-15-00608-f004:**
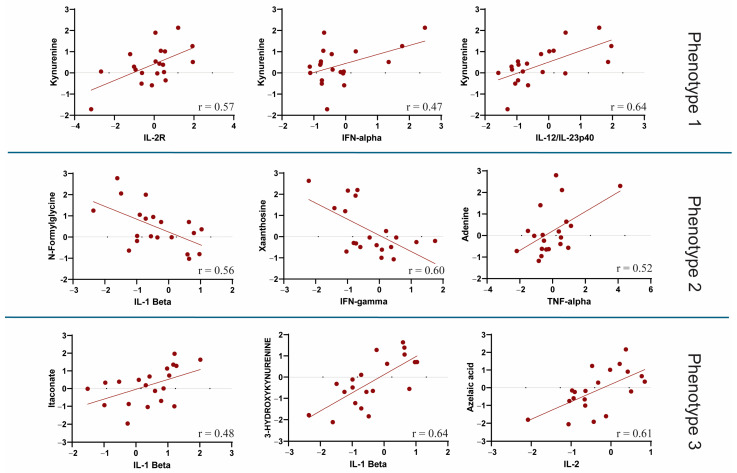
Examples of significant correlations (*p*-value < 0.05) between metabolites and cytokines among non-survivors are shown, except for itaconate and IL-1β, which were significantly correlated among survivors. The value of r represents the slope of the fit line.

**Figure 5 metabolites-15-00608-f005:**
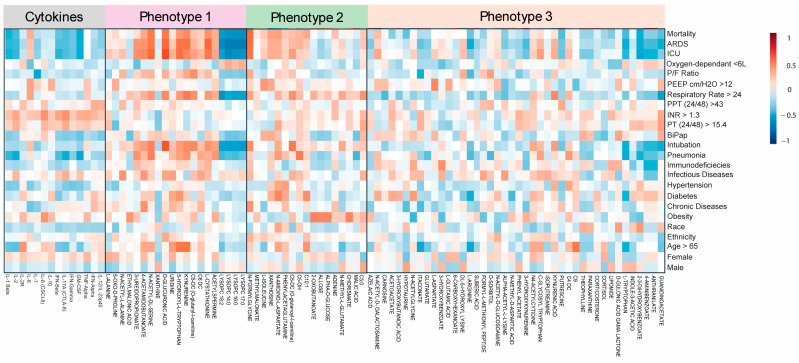
Correlations between cytokines, metabolites from metabolic phenotypes, and patients’ demographics, clinical data, and comorbidities.

**Table 1 metabolites-15-00608-t001:** Patient characteristics, clinical and comorbidities of two COVID-19 non-survivors (dead) and the COVID-19 alive cohorts.

Patients Characteristics	Alive (n = 20)	Death (n = 20)	*p*-Value
Sex (Male/Female)	10/10	10/10	1.000
Age	73 ± 11.4	70 ± 12.5	0.549
Ethnicity (Hispanic/Non-Hispanic)	16/4	18/2	0.428
Race (White/Black)	18/2	17/3	0.501
Obesity (Yes/No)	2/18	6/14	0.235
Chronic Diseases (Yes/No)	16/4	16/4	1.000
Diabetes (Yes/No)	3/17	7/13	0.137
Hypertension (Yes/No)	56/48	28/11	0.039
Infectious Diseases (Yes/No)	13/93	4/35	0.497
Immunodeficiencies (Yes/No)	15/92	8/31	0.239
Pneumonia (Yes/No)	13/7	19/1	0.055
Intubation/endotracheal tubes (Yes/No)	5/15	14/6	<0.0001
Bipap (Yes/No)	4/16	8/12	0.011
PT (24/48 after COVID infection)	15.2 ± 2.1	16.2 ± 5.6	0.491
PPT (24/48 after COVID infection)	35.7 ± 10.3	35.2 ± 9.7	0.904
INR	1.2 ± 0.20	1.32 ± 0.64	0.475
Respiratory rate	18.7 ± 4.2	26.2 ± 6.6	<0.0001
Tidal volume	357.5 ± 60.1	405.5 ± 39.3	0.149
PEEP cm/H_2_O	11.0 ± 1.41	11.8 ± 2.7	0.682
PaO2	113.0 ± 59.3	87.5 ± 43.5	0.226
FiO2	56.0 ± 30.7	75.3 ± 25.9	0.105
PF-ratio	247.3 ± 141.0	135.0 ± 79.0	0.016
*P*/*F* ratio (<100, 100–200, >201)	2, 1, 4	9, 5, 4	0.002
Oxygen-dependent (<6 L O_2_) (Yes/No)	10/10	4/16	0.006
ICU admission (Yes/No)	3/17	19/1	<0.0001
ARDS (Yes/No)	3/17	19/1	<0.0001
ICU length of stay	3.00 ± 3.01	16.8 ± 15.1	0.182
Length of stay to discharge or death	11.2 ± 8.7	16.5 ± 11.6	0.123

## Data Availability

The original contributions presented in this study are included in the article/Appendix A. Further inquiries can be directed to the corresponding author.

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
