# Peer review of "Metabolomics and Cytokine Signatures in COVID-19: Uncovering Immunometabolism in Pathogenesis"

_metabolites, 2025, doi:10.3390/metabo15090608_

Round 1

Reviewer 1 Report

Comments and Suggestions for Authors

The authors analyzed metabolic changes in blood samples from COVID-19 survivors and non-survivors. Some comments are provided below:

Introduction paragraph: 

1. “some key questions associated with the biochemistry of SARS-CoV-2
infections and inflammatory mechanisms remain unanswered.” It is not clear which questions exactly remain unanswered? The sentence needs to be reformulated.

2. “The elaborate interaction between the host immune response and metabolic changes partly explains the mechanisms behind the cytokine storm and multiorgan dysfunction in severe cases.” This does not explain the mechanism. Please provide more details.

3. “Significantly higher levels of inflammatory cytokines and inflammatory
markers, such as IL-6, IL-8, IL-10, and C-reactive protein and lactate dehydrogenase
(LDH), are associated with the critical group than with the moderate group.” There are many studies on this topic in the literature. Please add a review of the studies on this topic with links to relevant articles.

4. “Metabolic programming is a hallmark of immune cell activation that controls effector functions and fate decisions.” This is not clear. Please describe in more detail, provide references.

5. “The severity of the disease is attributed to the significant metabolic changes reported in COVID-19.” There are no references to support this statement.

6. Please add a review of studies on changes in carbohydrate metabolism, aminotransferase activity, lipid metabolism, and amino acid metabolism in severe cases of COVID-19.

7. “We aimed to determine the serum metabolomic profile of patients with COVID-19, including survivors and non-survivors, and their correlation with the cytokine profiles.” Why do you think that changes in the metabolic profile are associated with cytokine levels? What is the meaning of this correlation?

8. Paragraph “Cytokine profiling of COVID patients.” Please provide a figure demonstrating the cytokine profiles of surviving and deceased patients. “IgG levels” please clarify what IgG are we talking about?

9. Paragraph “Metabolomic profiling showed three main metabolic phenotypes in patients with COVID-19 compared with those in HCs”. Please provide a chromatogram, indicate the different metabolites. Describe the differences in metabolites in more detail. 

What is phenotype 1, 2, 3? How were patients assigned to one of the phenotypes?

10. Paragraph “Discussion”. What do the identified correlations indicate? What practical significance can this have in medicine?

Author Response

We thank the Reviewer for their thoughtful and constructive feedback. Your comments substantially improved the clarity, rigour, and clinical relevance of our manuscript. We have addressed each point in detail—clarifying aims and terminology, expanding mechanistic context, emphasizing associations rather than causation, strengthening the limitations, refining figures/supplementary materials, and updating references where appropriate. We appreciate the time and expertise invested in this review process.

  1. “some key questions associated with the biochemistry of SARS-CoV-2
    infections and inflammatory mechanisms remain unanswered.” It is not clear which questions exactly remain unanswered? The sentence needs to be reformulated.

Answer: Thank you for pointing out that the phrasing was too nonspecific. We have revised the manuscript to explicitly list major open questions in SARS‑CoV‑2 immunometabolism and inflammatory mechanisms. Below are key unresolved issues, now supported with recent literature. We add the following information to the manuscript:

Although research into COVID-19 immunometabolism has expanded, several central questions remain unanswered. First, the causal links and cellular origins of cytokine–metabolite networks such as the tryptophan/kynurenine and arginine pathways are still unclear (1, 2). Second, it is not well established how these networks shift from the acute stage of disease into recovery or Long COVID, or whether immune-directed therapies normalize the disrupted metabolism (1, 3). Third, the influence of obesity and related metabolic conditions on inflammatory metabolism needs clearer mechanistic explanation (4). Fourth, the extent to which viral variants and vaccination shape distinctive immunometabolic patterns is uncertain. Finally, it remains to be seen if combined cytokine and metabolite signatures can reliably predict outcomes across independent cohorts (5).

  1. E Maggi et al, (J Allergy Clin Immunl, 2020) COVID-19: Unanswered questions on immune response and pathogenesis.
  2. Xiao N, et al., (nature communication, 2021), Integrated cytokine and metabolite analysis reveals immunometabolic reprogramming in COVID-19 patients with therapeutic implications.
  3. R J Boyton et al. ( Nature Reviews Immunology, 2021) The immunology of asymptomatic SARS-CoV-2 infection: what are the key questions?
  4. Stefan et al. (Nature review endocrinology, 2012) Global pandemics interconnected — obesity, impaired metabolic health and COVID-19.
  5. H Mangge et al. (Biomedicines, 2021), Immune Responses against SARS-CoV-2—Questions and Experiences

  1. “The elaborate interaction between the host immune response and metabolic changes partly explains the mechanisms behind the cytokine storm and multiorgan dysfunction in severe cases.” This does not explain the mechanism. Please provide more details.

Answer:Thank you for your comments. We provided the details and added to the introduction.

Severe COVID-19 cytokine storm develops through a mix of immune over-activation, altered metabolism, and breakdown of normal control mechanisms. When viral components are detected by pattern recognition receptors such as TLRs, macrophages and dendritic cells release IL-6, TNF-α, and IL-1β. Instead of settling down, these signals are amplified by PAMPs, DAMPs, and inflammatory cell death, like pyroptosis or necroptosis (1). The result is a self-reinforcing cycle that drives widespread inflammation. High levels of cytokines then weaken endothelial and epithelial barriers, raise vascular permeability, trigger clotting pathways, and promote microthrombi. Damage spreads across the lungs, heart, kidneys, liver, and brain (2). Metabolic rewiring is deeply involved. Indoleamine 2,3-dioxygenase (IDO1) activation drains tryptophan and builds up kynurenine, which dampens T-cell activity but fuels AhR-driven pro-inflammatory signaling. Disturbed arginine metabolism limits nitric oxide, undermining endothelial function and encouraging clot formation. A glycolytic shift in immune cells, the Warburg effect, boosts lactate production and feeds back into cytokine release (3,4). Alongside this, depletion of glutathione, accumulation of ROS, and mitochondrial failure intensify tissue injury. When regulatory brakes like IL-10 signaling fail, the immune system cannot contract, and unchecked inflammation advances toward multi-organ failure (5).

  1. Vabret N, et al. Immunology of COVID-19: current state of the science. Immunity. 2020.
  2. Merad M, Martin JC. Pathological inflammation in patients with COVID-19. Nat Rev Immunol. 2020.
  3. Codo AC, et al. Elevated glucose fuels SARS-CoV-2 infection and monocyte response through a HIF-1α/glycolysis-dependent axis. Cell Metab. 2020.
  4. Thomas T, et al. COVID-19 infection alters kynurenine and fatty acid metabolism, correlating with IL-6 levels and renal status. Cell Rep Med. 2020.
  5. Mehta P, et al. COVID-19: consider cytokine storm syndromes. Lancet. 2020.

  1. “Significantly higher levels of inflammatory cytokines and inflammatory
    markers, such as IL-6, IL-8, IL-10, and C-reactive protein and lactate dehydrogenase
    (LDH), are associated with the critical group than with the moderate group.” There are many studies on this topic in the literature. Please add a review of the studies on this topic with links to relevant articles.

Answer:: Thank you for your comment: We have added a short review showing that IL-6, IL-8, IL-10, CRP, and LDH are consistently higher in severe/critical COVID-19 and associate with adverse outcomes.

Multiple cohorts and meta-analyses show that circulating cytokines and inflammatory markers consistently rise with worsening COVID-19. Elevated IL-6, IL-8, and IL-10 at the time of admission predict progression, ICU transfer, and mortality, even when adjusted for other variables (1). CRP is also markedly higher in severe cases compared to moderate disease and shows strong predictive value for adverse outcomes in both individual studies and quantitative reviews (2). LDH follows the same pattern, linking closely with severity, ICU requirement, and risk of death, with meta-analyses confirming its rise in severe or critical groups (3). Taken together, these findings align with our observation that IL-6, IL-8, IL-10, CRP, and LDH are enriched in critical illness and offer prognostic insight beyond the initial clinical presentation. (4)

  1. Hu H et al., Increased Circulating Cytokines Have a Role in COVID-19 Severity and Death With a More Pronounced Effect in Males: A Systematic Review and Meta-Analysis, Front Pharmacol, 2022.
  2. Asghar MS, et al. Poor Prognostic Biochemical Markers Predicting Fatalities Caused by COVID-19: A Retrospective Observational Study From a Developing Country, Curues 2020.
  3. Fiałek B et al., Diagnostic value of lactate dehydrogenase in COVID-19: A systematic review and meta-analysis, Cardiology Journal, 2020.
  4. Del Valle DM et al., An inflammatory cytokine signature predicts COVID-19 severity and survival. Nature Medicine, 2020.

  1. “Metabolic programming is a hallmark of immune cell activation that controls effector functions and fate decisions.” This is not clear. Please describe in more detail, provide references.

Answer:Thank you for your comment: We agree and have expanded the statement to clarify how metabolic programming controls immune effector functions and fate. We added the following review to the manuscript:

Environmental signals such as antigens, cytokines, hypoxia, and nutrients act through mTOR, HIF-1α, and AMPK to reprogram glycolysis, the TCA cycle, fatty acid oxidation, and oxidative phosphorylation, shaping immune cell activity. Naïve T cells depend on oxidative phosphorylation and fatty acid oxidation but shift to aerobic glycolysis after activation to sustain biomass growth and cytokine release. Memory T cells return to mitochondrial metabolism and fatty acid oxidation (1, 2). Lineage specification is also guided by metabolism: Th17 cells rely on glycolysis and glutaminolysis under HIF-1α and Myc control, while Tregs depend on fatty acid oxidation and mitochondrial pathways (1). In myeloid cells, M1 macrophages are glycolytic and accumulate succinate, which stabilizes HIF-1α to promote IL-1β. In contrast, the metabolite itaconate, produced by IRG1/ACOD1, limits inflammation by inhibiting SDH and activating Nrf2 (3-5). In COVID-19, monocytes display a HIF-1α–driven glycolytic shift with elevated lactate that amplifies cytokine production (6). Plasma kynurenine from IDO1-mediated tryptophan breakdown correlates with IL-6 levels and disease severity (7). Disturbed arginine and nitric oxide metabolism further disrupt endothelial function and host defense (8). These metabolic programs directly regulate cytokine production, antigen presentation, proliferation, cytotoxicity, and lineage fate rather than serving only as correlates (1).

  1. Buck MD, Sowell RT, Kaech SM, Pearce EL. Metabolic instruction of immunity. Cell (2017). Open-access summary.
  2. O’Neill LAJ, Pearce EJ. Immunometabolism governs dendritic cell and macrophage function. J Exp Med. 2016;213(1):15–23.
  3. Tannahill GM, et al. Succinate is an inflammatory signal that induces IL-1β through HIF-1α. 2013;496:238–242.
  4. Mills EL, et al. Itaconate is an anti-inflammatory metabolite that activates Nrf2 via alkylation of KEAP1. 2018;556:113–117.
  5. Peace CG, O’Neill LAJ. The role of itaconate in host defense and inflammation. J Clin Invest. 2022;132(2):e148548.
  6. Codo AC, et al. Elevated glucose levels favor SARS-CoV-2 infection and monocyte response through a HIF-1α/glycolysis-dependent axis. Cell Metab. 2020;32(3):437–446.e5.
  7. Thomas T, et al. COVID-19 infection alters kynurenine and fatty acid metabolism, correlating with IL-6 levels and renal status. JCI Insight. 2020;5(14):e140327.
  8. Durante W, et al. Targeting arginine in COVID-19–induced endothelial dysfunction and vasculopathy. Front Pharmacol. 2022;13:851756.

  1. “The severity of the disease is attributed to the significant metabolic changes reported in COVID-19.” There are no references to support this statement.

Answer:Thank you for your comment: We added these two references to support the statement;

Sindelar et al., Cell Reports Medicine (2021): Longitudinal plasma metabolomics early in disease course predicted subsequent severity, highlighting pathway-level remodeling.

Shen et al., Cell (2020) Proteomic and Metabolomic Characterization of COVID-19 Patient Sera

  1. Please add a review of studies on changes in carbohydrate metabolism, aminotransferase activity, lipid metabolism, and amino acid metabolism in severe cases of COVID-19.

Answer:Thank you for your comment: we have added the following review to the manuscript

Severe COVID-19 is marked by broad metabolic remodeling. Carbohydrate pathways shift toward hyper-glycolysis, with HIF-1α–driven monocyte reprogramming and increased lactate that fuels inflammatory signaling [1]. Aminotransferase activity often rises, and a recent systematic review and meta-analysis reported that both the De Ritis ratio (AST/ALT) and elevated AST or ALT are linked to greater severity and higher mortality [2]. Lipid handling is also disrupted, with altered transport and plasma lipid signatures that track with disease severity in cross-omic studies [3]. Amino acid metabolism changes as well: activation of the tryptophan to kynurenine pathway aligns with IL-6 levels and clinical status, while disrupted arginine and nitric oxide pathways impair endothelial and immune function [4,5]. Collectively, these alterations reveal a recurring pattern of metabolic rewiring that parallels escalating inflammation and poor outcomes..

  1. Codo AC, et al. Elevated glucose levels favor SARS-CoV-2 infection and monocyte response through a HIF-1α/glycolysis-dependent axis. Cell Metab.
  2. Mangoni AA, et al. An updated systematic review and meta-analysis of the De Ritis ratio in COVID-19: association with severity and mortality. Life (Basel). 2023;13(6):1324.
  3. Overmyer KA, et al. Large-scale multi-omic analysis of COVID-19 severity (transcripts, proteins, metabolites and lipids) identifies severity-tracking lipid and metabolic signatures. Cell Syst.
  4. Thomas T, et al. COVID-19 infection alters kynurenine and fatty-acid metabolism, correlating with IL-6 levels and renal status. JCI Insight. 2020;5(14):e140327.
  5. Durante W, et al. Targeting arginine/NO pathways in COVID-19–induced endothelial dysfunction and vasculopathy. Front Pharmacol. 2022;13:851756. PMC

  1. “We aimed to determine the serum metabolomic profile of patients with COVID-19, including survivors and non-survivors, and their correlation with the cytokine profiles.” Why do you think that changes in the metabolic profile are associated with cytokine levels? What is the meaning of this correlation?

Answer: Thank you for this helpful comment. We have clarified our rationale in the Introduction by continuing to describe the aims of the study as follows:

We hypothesized that these correlations would index a coordinated immunometabolic state because inflammatory cytokines actively reprogram metabolism and metabolites reciprocally modulate cytokine output; accordingly, we treat correlations as composite readouts of disease activity rather than causal claims.”

  1. Paragraph “Cytokine profiling of COVID patients.” Please provide a figure demonstrating the cytokine profiles of surviving and deceased patients. “IgG levels” please clarify what IgG are we talking about?

Answer: Thank you for your comment: we have added the following figure of the boxplots to the supplementary material to show the details of the cytokine profile among survivors and non-survivors.

Total serum immunoglobulin G (IgG) was quantified from admission serum using a validated clinical immunoassay and reported in g/L. IgG subclasses and other isotypes were not measured.”

  1. Paragraph “Metabolomic profiling showed three main metabolic phenotypes in patients with COVID-19 compared with those in HCs”. Please provide a chromatogram, indicate the different metabolites. Describe the differences in metabolites in more detail. 

Answer: Thank you for your comment: we described the differences in metabolites in more detail and added the three chromatograms of 3 examples from each phenotype as follows to the supplementary:

And we integrate the following description to the mentioned section:

Phenotype 1 — Non-survivor–enriched, inflammatory catabolism (Figure 2A–B).
This group of metabolites appears highest in non-survivors, lower in survivors, and lowest in healthy controls. Examples include kynurenine from the tryptophan–kynurenine pathway, acetylspermidine from polyamine turnover, and 5-hydroxytryptophan as a serotonin precursor. Boxplots in Figure 2B illustrate the stepwise rise across groups with exact p-values noted, and representative chromatograms are shown in Figure 5S. The pattern reflects IDO1/AhR-driven inflammatory catabolism along with enhanced polyamine acetylation in fatal cases.

Phenotype 2 — Energy and nucleotide remodeling (Figure 2A–B).
Carbohydrate and purine metabolism shift with disease severity. Glucose is most elevated in non-survivors, xanthosine from purine breakdown also rises in this group, while phenylacetylglutamine (PAGln) is comparatively higher in survivors. These findings point to hyper-glycolytic stress and accelerated nucleotide turnover in the poorest outcomes, with survivors showing a distinct gut–liver axis signal through PAGln. Boxplots and chromatograms are presented in Figure 2B and Figure 5S.

Phenotype 3 — Oxidative kynurenine branch and counter-regulation (Figure 2A–B).
This module links oxidative kynurenine pathway activity with immunoregulatory signals. 3-hydroxykynurenine is elevated in non-survivors, azelaic acid rises in both COVID groups compared with healthy controls, and itaconate, a macrophage-derived metabolite, is higher in COVID-19 with a trend toward greater levels in survivors, suggesting a more active counter-inflammatory response. Group differences are shown in the boxplots of Figure 2B, with chromatograms in Figure 5S.

What is phenotype 1, 2, 3? How were patients assigned to one of the phenotypes?

Answer: In our manuscript, “Phenotypes 1–3” refer to metabolic phenotypes (“metabopatterns”)—i.e., sets of metabolites that distinguish the three key group comparisons (non-survivors vs survivors; non-survivors vs healthy controls; survivors vs healthy controls). These phenotypes organize metabolites, not patients; each patient can exhibit contributions from all three. Figure 2 (heatmap) is arranged by these metabopatterns, with representative examples shown in the boxplots.

Patients are not restricted to one phenotype, since each sample can carry metabolites from all three patterns. The heatmap in Figure 2 is arranged according to these metabopatterns, and the boxplots highlight representative metabolites for each comparison.

  1. Paragraph “Discussion”. What do the identified correlations indicate? What practical significance can this have in medicine?

Answer: Thank you for your comment. We added this discussion bout our findings to the manuscript.

The correlations identified point to linked immune and metabolic activity rather than independent signals. Inflammatory cytokines such as IL-6 and IFN pathways can redirect metabolism toward routes like tryptophan to kynurenine or glycolysis to lactate, while certain metabolites in turn influence cytokine output and vascular or immune function. These paired readouts therefore capture integrated features of disease biology. Clinically, such signatures have several applications: they can sharpen prognostic stratification when added to standard labs and covariates, help define immunometabolic subtypes such as IL-6–kynurenine–driven versus glycolysis/lactate–driven patterns that may guide therapy or trial design and serve as markers of pharmacodynamic response to anti-inflammatory or supportive treatments. Although correlation does not establish causation, the biological logic of these axes makes them suitable for targeted validation in independent cohorts and longitudinal studies.

Reviewer 2 Report

Comments and Suggestions for Authors

1- The abstract is informative but could be streamlined to highlight the most clinically relevant findings.

2- The cohort is relatively small (40 patients and 40 controls), which limits statistical power and raises concerns about generalizability. Please discuss this limitation more explicitly in the manuscript.

3- The study focuses only on hospitalized moderate-to-severe COVID-19 patients. Mild or outpatient cases were not included, which restricts the interpretation of metabolic signatures across the full clinical spectrum. This should be acknowledged.

4- The enrollment period (2020–2021) coincided with different SARS-CoV-2 variants (Alpha, Delta). Since viral variants can modulate immune and metabolic responses, the lack of variant stratification weakens the generalizability of the findings. Please address this issue.

5- Samples were collected “within the first two days of illness,” but since patients were hospitalized at different stages of disease, the baseline may not be consistent. A clearer justification or discussion of this variability is needed.

6- PCA and PLS-DA were applied, with corrections (FDR, Bonferroni). However, model validation metrics (e.g., R², Q², cross-validation, or permutation tests) are not reported. Without these, the risk of model overfitting cannot be excluded. Please include these quality control measures.

7- While multiple correlations between metabolites and cytokines are reported, the manuscript often implies causality. It would be more appropriate to emphasize that these are associations and not necessarily causal relationships.

8- The observation of stronger cytokine–metabolite correlations in females is intriguing, but the biological explanation is underdeveloped. Please expand the discussion to include possible hormonal, genetic, or immune-response differences between sexes.

9- Although correlations with comorbidities (e.g., obesity, diabetes) are shown, the impact of potential confounders such as medications (steroids, antivirals) is not addressed. This is important and should be discussed.

10- Since all patients were recruited from a single hospital (University of Miami), the results may be influenced by specific demographic or regional factors. Please acknowledge this limitation.

11- The manuscript cites several relevant studies, but additional comparison with large-scale multi-omics studies published in Cell, Nature, or Nature Medicine (2020–2022) would place these findings in a stronger global context.

12- The conclusion mentions metabolites as potential biomarkers, but ROC curve data are not fully elaborated. Identifying which metabolites achieved the highest predictive performance (sensitivity, specificity, AUC) would substantially improve the translational value.

Author Response

We thank the Reviewer for their thoughtful and constructive feedback. Your comments substantially improved the clarity, rigour, and clinical relevance of our manuscript. We have addressed each point in detail—clarifying aims and terminology, expanding mechanistic context, emphasizing associations rather than causation, strengthening the limitations, refining figures/supplementary materials, and updating references where appropriate. We appreciate the time and expertise invested in this review process.

Review #2

  • The abstract is informative but could be streamlined to highlight the most clinically relevant findings.

Answer: Thank you for your comment we added the following relevant findings to the abstract that could be clinically relevant.

“Compared with survivors and healthy controls, non-survivors displayed higher IL-6 together with distinct metabolic changes. These included increased kynurenine through the IDO1 pathway, elevated glucose and lactate reflecting hyper-glycolysis and energy stress, and higher xanthosine from purine turnover. Stronger cytokine–metabolite correlations in this group point to tightly linked immunometabolic activation.

  • The cohort is relatively small (40 patients and 40 controls), which limits statistical power and raises concerns about generalizability. Please discuss this limitation more explicitly in the manuscript.

Answer: Thank you for your comment, we added this to the limitations in the discussion.

Also, this study was a modest single-centre cohort, both power and generalizability are limited. Even with FDR correction, some results may be unstable and will need confirmation in larger multi-centre studies.”

  • The study focuses only on hospitalized moderate-to-severe COVID-19 patients. Mild or outpatient cases were not included, which restricts the interpretation of metabolic signatures across the full clinical spectrum. This should be acknowledged.

Answer:Thank you for your comment, we added this to the limitations also and acknowledged it.

Our study population consisted only of hospitalized patients with moderate to severe COVID-19, as mild or outpatient cases were not included. The immunometabolic signatures we describe may therefore reflect inpatient disease biology and should not be assumed to apply to early or ambulatory infection. Cohorts of community-based mild or asymptomatic cases will be needed to test generalizability across the full spectrum of disease.

  • The enrollment period (2020–2021) coincided with different SARS-CoV-2 variants (Alpha, Delta). Since viral variants can modulate immune and metabolic responses, the lack of variant stratification weakens the generalizability of the findings. Please address this issue.

Answer: Thank you for your valuable comment, we add this to the before section of limitations of this work.

Patient enrollment (2020–2021) took place across several SARS-CoV-2 variant waves, including Alpha and Delta. Viral genotyping was not available, so analyses were not stratified by variant. Since variants differ in both immune and metabolic responses, and clinical management changed during this period, the applicability of our immunometabolic signatures to specific variant contexts—particularly the Omicron era—is limited. Future work should include viral sequencing or reliable proxy assignment and conduct variant-stratified analyses.”

  • Samples were collected “within the first two days of illness,” but since patients were hospitalized at different stages of disease, the baseline may not be consistent. A clearer justification or discussion of this variability is needed.

Answer:Thank you for noting this valuable comment: we added this discussion to the before  limitation of works to

Since patients were admitted at different points in their illness, our baseline reflects an admission-time snapshot rather than a consistent disease day. This heterogeneity may blur or weaken stage-specific effects. Studies that stratify by illness duration and include serial sampling will be important to confirm these observations.

  • PCA and PLS-DA were applied, with corrections (FDR, Bonferroni). However, model validation metrics (e.g., R², Q², cross-validation, or permutation tests) are not reported. Without these, the risk of model overfitting cannot be excluded. Please include these quality control measures.

Answer: Thank you for your comment, we agree with the lack of information. We added the R2 and Q2 and p values to the PLS-DA plots, and change the legend.

  • While multiple correlations between metabolites and cytokines are reported, the manuscript often implies causality. It would be more appropriate to emphasize that these are associations and not necessarily causal relationships.

Answer: Thank you for your valuable comment, we absolutely agree, as we went through the, we find it is better to emphasize this with a few words at the beginning of the discussion with these words:

The cytokine–metabolite relationships we report are associative rather than causal. We view these correlations as integrated markers of ongoing immunometabolic activity that may aid risk stratification, while acknowledging that causal links will need to be tested in longitudinal and perturbational studies.

  • The observation of stronger cytokine–metabolite correlations in females is intriguing, but the biological explanation is underdeveloped. Please expand the discussion to include possible hormonal, genetic, or immune-response differences between sexes.

Answer: Thank you for this very important comment. We added this information with the references to explain this issue:

We found that cytokine–metabolite correlations were stronger in females. This is biologically plausible for several reasons. Hormonal influences play a role: estrogens tend to boost innate antiviral and type I IFN responses as well as humoral immunity, while androgens can shape ACE2/TMPRSS2 expression and inflammatory pathways. X-linked immune genes such as TLR7 may also contribute, since gene-dose or escape effects can heighten IFN signaling in some women. In addition, sex-related metabolic remodeling has been reported in COVID-19, spanning lipid, pentose, bile acid, and aromatic amino acid/tryptophan pathways. Prior studies further show sex-dependent differences in the kynurenine axis, including distinct associations of kynurenic acid by sex, which emphasizes that immunometabolic coupling does not operate identically in males and females. We interpret the stronger correlations in females as associative evidence of tighter immunometabolic coordination, while recognizing the limitations of sample size, treatment and variant heterogeneity, and the need for independent validation (1-3).

  1. Ciarambion T et al. Women Health Journal,2021, Immune system and COVID-19 by sex differences and age.
  2. Naz M. et al Future Virology, 2021, An overview of sex hormones in relation to SARS-CoV-2 infection.
  3. Escarcega R et al, nature 2022, Sex differences in global metabolomic profiles of COVID-19 patients

9- Although correlations with comorbidities (e.g., obesity, diabetes) are shown, the impact of potential confounders such as medications (steroids, antivirals) is not addressed. This is important and should be discussed.

Answer: Thank you for your comment: we agree with this statement, and we added the text to the discussion as follows:

In-hospital therapies such as glucocorticoids, antivirals, IL-6 receptor blockers, anticoagulants, and glucose-lowering agents can rapidly influence cytokine and metabolic profiles. Detailed information on timing and dosing at the time of sampling was not consistently available, and the cohort size did not allow for robust adjustment. Residual confounding is therefore likely. The profiles presented here should be viewed as reflecting both disease activity and concurrent treatment, and future work will need to include more granular medication data and stratified analyses.”

  • Since all patients were recruited from a single hospital (University of Miami), the results may be influenced by specific demographic or regional factors. Please acknowledge this limitation.

Answer: Thank you for your comment, we have already mentioned one our answer to your comment above to address that a single center for collecting samples may not represent the generality

  • The manuscript cites several relevant studies, but additional comparison with large-scale multi-omics studies published in CellNature, or Nature Medicine(2020–2022) would place these findings in a stronger global context.

Answer: Thank you for your comment. During the revision of this paper based on the reviewers' comments, we have already used several papers that have been published in Nature and Cell or high-impact journals that provide relevant information to the findings.

  • The conclusion mentions metabolites as potential biomarkers, but ROC curve data are not fully elaborated. Identifying which metabolites achieved the highest predictive performance (sensitivity, specificity, AUC) would substantially improve the translational value.

Answer: Thank you for this constructive comment. Our primary aim in this study was to define metabolic phenotypes (metabopatterns) based on how metabolites change across non-survivors, survivors, and healthy controls, and to examine their correlations with cytokines—not to rank individual metabolites as standalone diagnostics. We therefore did not compute per-metabolite ROC, sensitivity, or specificity in the main analysis.

We apologize that we are unable to provide robust per-metabolite ROC estimates in this cohort. We have tempered our conclusions accordingly and now frame these metabolites as candidate markers requiring validation. A companion study focused on diagnostic/prognostic performance—with detailed per-metabolite and multivariable ROC metrics—is underway and will report sensitivity, specificity, and AUC with appropriate validation. We corrected the methods and materials for clarity.

Round 2

Reviewer 1 Report

Comments and Suggestions for Authors

The new additions to the manuscript made a big difference. The quality of the paper had improved, and all my questions were addressed. No more comments.

Reviewer 2 Report

Comments and Suggestions for Authors

Thanks